# On the Construct of Functional Psychology's Developmental Theory: Basic Experiences of the Self (BEsS)

Filippo Dipasquale [1,*], Marta Blandini [2], Raffaele Gueli [2], Paola Fecarotta [2] and Paola Magnano [3]

1   Independent Researcher, 95127 Catania, Italy
2   Functional Psychotherapy Centre "Wilhelm Reich", 95124 Catania, Italy; martablandini@gmail.com (M.B.); gueliraffaele.psi@gmail.com (R.G.); paola.fecarotta@tiscali.it (P.F.)
3   Faculty of Human and Social Sciences, Kore University of Enna, 94100 Enna, Italy; paola.magnano@unikore.it
*   Correspondence: f.dipas@gmail.com

**Abstract:** According to the neo-functional developmental theory, newborns and infants exhibit complex psycho-bodily functioning. The Basic Experiences of the Self (BEsS) refer to how they fulfil their essential life needs by organising their psycho-bodily functions in a typical configuration. As part of our research study, we developed a prototype psychometric tool called the BEsS Assessment Form (BAF) to assess the BEsS in infants aged zero to three years. We collected video recordings of their spontaneous behaviour and used the BAF to evaluate function polarity. In the BAF, thirty pairs of words represent functions in their dyadic polarity. To estimate the level of function polarity, we used the Osgood semantic differential scale, which ranges from seven to one. The study's results confirm that functions can be assessed by grading along the opposite polarity spectrum. Moreover, in accordance with the theory, the functions can be grouped into four domains: the emotional, postural motor, physiological, and cognitive-symbolic planes. Our findings suggest that the characteristics of BEsS are significantly influenced by the activation of the physiological and postural motor functions, which are related to the early regulation of the autonomic nervous system and can be used to evaluate infant arousal.

**Keywords:** basic experiences of the self; infant development; first one thousand days of life; network analysis; DOHaD; autonomic nervous system (ANS) autoregulation

## 1. Introduction

Developmental plasticity refers to the capability of human biology to adapt to various environments [1]. It plays a crucial role in the first one thousand days of life development, from conception through the first two years of life. Early experiences can have a lifelong effect on individual health and well-being by 'programming' organs, tissues, or body system structures and functions. They can even influence health by increasing the risk of chronic diseases [2,3], according to the Developmental Origins of Health and Disease (DOHaD) hypothesis [4], which represents a modern epidemiological point of view that has been driving international policymaking for cures and preventive health strategies [5,6].

A suitable environment is crucial for typical neurodevelopment during early childhood, and the risk factors are crucial for neurodevelopmental disorders [7,8]. It is important to understand that referring to brain development in terms of building neuronal connections or brain architecture is an oversimplification of the complex functioning of the brain. The brain is not isolated but closely linked with other vital bodily systems. These systems, such as the nervous, endocrine, and immune systems, are finely integrated, meaning that the mind and body are inherently interconnected [9].

Child development involves more than motor or cognitive skills, as learning is not always a conscious process, especially in early development. In a typical child development trajectory, every observable behaviour represents a functional and integrated process

which involves various aspects, such as motor skills, cognitive abilities, emotions, and physiological responses. It is time for clinical practitioners to consider observable clinical findings in children as being the result of integrated systems functioning. This will help to align clinical practice with developmental theory, which acknowledges the complexity and variability [10]. It should provide scientific evidence to validate the constructs used in clinical practice.

## 1.1. Psycho-Bodily Significance in Neo-Functionalism

The significance of psycho-bodily effects in treating psychic dysfunctions dates back to the time of Pierre Janet, who introduced his theory on hysteria at the Amsterdam conference in 1907. In 1997, Boadella [11] emphasised the importance of incorporating massage and working with the body in the treatment of trauma patients, citing specific elements such as the diaphragmatic block and visceral consciousness. The potential of psychotherapy to modify brain function and impact epigenetic aspects in the long term has been extensively discussed, much like other forms of treatment, such as pharmacology [12]. Psychotherapy commonly focuses on complex somatic disorders caused by psychic conflicts. Emphasis is placed on the comprehensive relationship between the mind and body, as there are various physical effects that can result from psychological factors [13,14]. Recent studies in neuroscience have confirmed and highlighted the significance of bodily integrated expressions in the relationship between the body and mind, affirming a growing interest in the role that the body plays in functional somatic disorders [15]. This is a focus that is viewed as a complex issue in clinical practice and one that can be challenging to treat. As a result, integrative treatment strategies are often necessary.

Neo-functionalism in psychology is based on the functionalism of the Chicago School (Dewey, Angell, and Carr, 1905–1925) and builds on Darwinian evolutionism. The mind–body connection requires a new concept that integrates their functioning. The importance of the body in psychotherapy has been a recurring theme for almost a century. Fluctuations in the recognition of the body's significance in psychotherapy have occurred over time, with periods of neglect followed by re-emergence, as evidenced by the neo-Reichian approach. Functional body psychotherapy was developed at Rispoli's Naples school, drawing inspiration from Wilhelm Reich's theories [16,17]. For many years, empirical and practical studies have been conducted on neo-functionalism, which suggests that integrating our physical and psychological experiences leads to the development of the self [18]. The self-concept refers to the set of beliefs we have about ourselves. According to functional theory, this process should begin in the early stages of life and should result in a complex and functional organisation with the development of an original self-concept. The self is the outcome of life experiences and represents the result of our complex functioning shaped during development. Our awareness of physical and psychological characteristics depends on functioning during life experiences. Throughout our lifespan, social roles, values, goals, and memories continue to shape our sense of self. This concept is a significant advancement in developmental theory.

Today, functional psychotherapy is an integrated approach that considers psychological dysfunctions as complex embodied disorders. It employs methodological strategies for diagnosis and therapeutic techniques to achieve psycho-bodily integration. The functional method aims to achieve overall physical and mental well-being by addressing the effects on the body and the related emotional and psychological conflicts. Recent empirical studies examined the efficacy of functional psychotherapy, introducing an innovative concept in clinical psychology. Based on evidence from neuroscience, it is considered an epistemological breakthrough in the research [19–24].

## 1.2. The Neo-Functionalism Developmental Theory

Neo-functionalism focuses its developmental theory on the Basic Experience of the Self (BES) as the foundational construct. The Basic Experiences of the Self (BEsS) represent the infancy starting point for the complex psycho-bodily functioning of adulthood.

According to this concept, children fulfil their specific needs, such as contact, feeding, and being held through the Basic Experiences of the Self (BEsS) that occur from birth and continue throughout their development. Each BES can be seen as a fundamental building block of an individual's self and represents typical and recognisable life events. In line with the theory, the self develops through BEsS in one's life, gradually becoming more intricate and serving as the foundation for an individual's typical functioning in adulthood. Empirically, numerous BEsS have been accurately defined and described in terms of their functional psycho-bodily qualities in clinical practice. The BEsS express their typical characteristics through the organisation of psycho-bodily functions, the smallest and fundamental features in a child's complex functioning. Each function is assigned to a virtual functional psycho-bodily plane: the physiological, postural motor, emotional, and cognitive-symbolic. The neo-functionalism theoretical model proposes that psycho-bodily functions replace parts as essential functional units. These functions do not only refer to physiological processes involving specific organs, as conventional medicine suggests; they also result from the close integration of psychological, biological, and physiological factors at all organism levels [17,18]. According to functional theory, all psycho-bodily functions are innate and exist on a spectrum between two opposite poles. For example, happiness and sadness are polarities of the same range, like active and passive. During development, a child's needs are met through their BEsS, and all psycho-bodily functions move fluently between their polarities, which shapes their typical lived experiences. This complex fine-tuning of functions and their organisation during development becomes the foundation of adult functioning.

## 2. Rationale of the Study

This study analyses 16 BEsS described in Rispoli's work [18] (p. 48) and follows up on a previous study, to which we refer for a deeper explanation of the functionalist model and methodological research design [25].

1. Vitality is an energy that enhances both the inner and outer self with joy and energy.
2. Contact is a flux of sensations continuously flowing from one to another, encompassing nearness, fusion, or empathy.
3. Letting Go means entering a state of rest, loosening the state of activation or enchantment after activity.
4. Being Held refers to the experience of being firmly embraced and feeling warmth and protection.
5. Tenderness is a child's gentle approach, behaviour, and openness to others.
6. Strength refers to the experience of asserting oneself clearly and openly, firmly but quietly.
7. Joyous Aggression refers to the tendency to channel one's strength against others playfully and joyously.
8. Loosening Control is the ability to relax after reducing control that has overloaded the body and mind.
9. Playing is essential for children to explore their environment with joy and pleasure.
10. Just Being is the child's way of remaining inactive, abstaining from any act. It is a natural consequence of letting go and embracing one's authentic self.
11. Calmness forms the foundation of patience, tranquillity, and a serene perception of life experiences.
12. Autonomy is a BES with which to relate to others consistently; it means being comfortable with oneself while still connected to others.
13. Affirmation is a child's way of creating their own space and being recognised as they strive to achieve social goals.
14. Changing Others: From birth, infants exhibit remarkable strength in moving those around them, demonstrating the potential for change in others.
15. Pleasure: Experiencing pleasure is a profoundly satisfying feeling that arises from creating something and simply existing.
16. Opposition is a way of expressing disapproval or disagreement.

During development, a child undergoes various experiences known as BEsS [17,18]. This process helps in the formation of their sense of self through repeated experience and the adjusting of all their functions to contribute equally to their overall functioning, in line with the diverse environmental situations they encounter. This rich exposure to life situations helps the child develop congruence with their environment from the early stages of life, especially within the first one thousand days. Each person's sensory system processes experiences differently, leading to a unique and consistent personal experience. This is based on how psycho-bodily functions are organised in each BES, depending on the subtle nuances and varying levels of intensity through the spectrum in their opposite polarity. The more congruently and fluidly a child lives BEsS, the more mobile and adequate are the changes in the configuration of the functions according to their functioning.

In a previous study [25], researchers developed a psychometric tool called the BAF—BES Assessment Form (shown in Appendix A)—to evaluate the activity of children recorded on video. The study focused on the BEsS, including Vitality, Contact, and Letting Go, and on analysing videos and checking the polarity of their functions. A total of 840 BAFs were collected.

Our research is focused on validating the constructs related to neo-functionalism. According to the functionalism hypothesis of the developmental theory, each BES has a characteristic configuration of functions, which can be quantitatively evaluated by grading the functions based on their polarity. Additionally, these functions can be classified into four domains. Our goal is to prove that by quantitatively assessing the functions and their polarity, we can identify the unique characteristics of each BES.

### 3. Materials and Methods

*3.1. Data Collection*

A collection of short video clips showing the spontaneous activity of infants aged from zero to three years was available. The families of the students at the European Psychotherapy School provided these clips, and the children's parents gave informed consent for inclusion before participating in the study. The study was conducted in accordance with the Declaration of Helsinki, and the protocol was approved by the Institutional Review Board of the Functional Psychotherapy Centre, Catania, Italy. (Protocol code 39/2022. Date of approval: 14 November 2022.)

The authors chose videos based on previously established criteria [25]. Infants of both sexes, aged between zero and three years, with no history or evidence of neurodevelopmental risks or disorders, were invited to participate in the study. Most of the infants' videos were selected through convenience sampling from the archives of the families associated with the students of the European School of Functional Psychotherapy. The videos depict various life scenes, and environmental conditions did not significantly affect the BEsS characteristics. A group of expert psychotherapists and students from the European School of Functional Psychotherapy with degrees in psychology or medicine evaluated the child's recorded activities. They assessed thirty psycho-physical functions on the Basic Experiences of the Self Assessment Form (BAF), rating them according to opposite dyadic polarities. The functions were translated into English with minimal word changes to reduce the semantic understanding bias and to eliminate redundancies. We used Osgood's semantic differential scale [26] to determine the degree of polarity associated with each function. The scale assigns a score from 7 to 5 for the level of the left-side polarity, where 7 means "very much", 6 means "enough", and 5 means "a little". Similarly, a score from 3 to 1 is assigned for the level of polarity on the right side, with 3 representing "a little", 2 representing "enough", and 1 representing "very much". A score of 4 is considered neutral on the scale.

We followed the same criteria for data analysis as the previous study [25], which included forms with no more than three missing functions and accepted a score of 4 for the correction. Only one value was allowed per line. Table 1 shows the number of collected forms and the assessed videos for any BES.

**Table 1.** Collected BAFs admitted to the study. Numbers of collected videos and number of BAFs admitted to the study. Ordered form numbers vary by BES due to differences in video and participant counts.

|  | Videos n. | BAF n. |
|---|---|---|
| Vitality | 10 | 316 |
| Contact | 9 | 256 |
| Letting Go | 9 | 268 |
| Being Held | 7 | 211 |
| Tenderness | 8 | 373 |
| Strength | 6 | 288 |
| Joyous Aggression | 5 | 195 |
| Loosening Control | 7 | 267 |
| Playing | 9 | 349 |
| Just Being | 8 | 317 |
| Calmness | 15 | 530 |
| Autonomy | 15 | 618 |
| Affirmation | 16 | 595 |
| Changing Others | 11 | 365 |
| Pleasure | 11 | 405 |
| Opposition | 11 | 278 |
| Total | 157 | 5631 |

*3.2. Data Analysis*

We employed the JASP computer software (Version 0.17.2) developed by Team (2023) and IBM SPSS Statistic software for Mac (Ver. 26.0.0, released in 2019) to perform statistical data analysis. These software applications provide an extensive range of analytical instruments.

To assess the internal consistency of the BAF for any BES, we conducted a Cronbach's alpha test on the collected data. After that, we employed network analysis to examine the complex configuration of the functions within BEsS. Network analysis is a valuable technique used in psychology for analysing multivariate data, and it is particularly useful for exploratory analysis, especially when sparse network structures are used to better illustrate organisational insights [27,28]. We examined the interrelations between the functions using network analysis and evaluated how they correlated with the physiological, postural motor, emotional, and cognitive-symbolic domains.

Our previous research [25] discovered that two out of three BEsS exhibit a distinct polarity inversion in specific physiological and postural motor functions. This suggests that there could be a reduction in physiological activation in observed child behaviour. In this study, we further analysed this trait in other BEsS. We used ANOVA tests with post hoc Bonferroni and Tukey analyses, Welch's *t*-test for independent samples, and Cohen's d-effect size measurements to investigate variations among BEsS for these functions.

**4. Results**

This study's Cronbach's alpha results from the BAFs are presented in Table 2. The results indicate that the internal consistency is good and acceptable ($\alpha > 0.80$ and $\alpha > 0.70$, respectively) for all but two of the BEsS. The BES Letting Go ($\alpha = 0.634$) and the BES Being Held ($\alpha = 0.696$) showed questionable internal consistency. Overall, the results confirm the good internal consistency of the BAF instrument.

The functions can be classified into four domains: physiological, emotional, postural motor, and cognitive-symbolic [17,18]. To confirm this classification, network analysis was adopted. In network analysis, the nodes represent variables, while the edges symbolise the association between two nodes. A bidirectional correlation represents their conditional associations. As more associations reach the nodes, their correlation changes over time. A network approach to multivariate data involves estimating a network structure, describing the network, and confirming the network's stability and robustness. It is a helpful method for representing a model factor [27]. To appropriately represent functions in network

analysis, each function was first labelled, as listed in Table 3. We explored network structure estimation and used a partial correlation option to compute the network organisation by applying the method of high node sparsity [28].

**Table 2.** Internal consistency for each studied BES. Ten out of the sixteen BEsS showed good Cronbach's alpha values; three were acceptable; and two were questionable.

|  | Cronbach's $\alpha$ |
|---|---|
| Vitality | 0.845 |
| Contact | 0.783 |
| Letting Go | 0.634 |
| Being Held | 0.696 |
| Tenderness | 0.791 |
| Strength | 0.860 |
| Joyous Aggression | 0.854 |
| Loosening Control | 0.804 |
| Playing | 0.862 |
| Just Being | 0.769 |
| Calmness | 0.821 |
| Autonomy | 0.862 |
| Affirmation | 0.860 |
| Changing Others | 0.854 |
| Pleasure | 0.828 |
| Opposition | 0.858 |

**Table 3.** List of thirty studied functions. The labels represent the grouping functions in functional planes: E, emotional; PM, postural motor; PH, physiological; CS, cognitive-symbolic.

| Functions | Label |
|---|---|
| Certainty/Uncertainty | E1 |
| Happiness/Unhappiness | E2 |
| Tranquillity/Agitation | E3 |
| Hope/Hopelessness | E4 |
| Trust/Distrust | E5 |
| Calm/Anger | E6 |
| Love/Hate | E7 |
| Mobility/Immobility | PM1 |
| Fast Movements/Slow Movements | PM2 |
| Active/Passive | PM3 |
| Nimble/Clumsy | PM4 |
| Light/Heavy | PM5 |
| Big Movements/Small Movements | PM6 |
| Soft Movements/Jerky Movements | PM7 |
| Softness/Stiffness | PM8 |
| Harmonious Body/Disharmonious Body | PM9 |
| Open Voice/Closed Voice | PH1 |
| Excitation/Tranquillity | PH2 |
| Noisy/Silent | PH3 |
| Sympathicotonia/Vagotonia | PH4 |
| Sensibility/Insensibility | PH5 |
| Muscle Hypertonia/Muscle Hypotony | PH6 |
| Thoracic Breathing/Diaphragmatic Breathing | PH7 |
| Attention/Loosening | CS1 |
| Rational/Irrational | CS2 |
| Fantasy/Concreteness | CS3 |
| Remember/Forget | CS4 |
| Positive Values/Negative Values | CS5 |
| Creative Thinking/Logical Thinking | CS6 |
| Globality/Details | CS7 |

The network analysis illustrated in Figure 1 displays the organisation of nodes into the four functional planes through the graphical aggregation of labelled functions. We used the estimated partial correlation and set a threshold that excluded edges with values less than 0.08. This produced a high-sparsity structure that emphasised intragroup edge density over intergroup areas. The results confirmed the theory by showing the betweenness and closeness relations of the emotional nodes (green), postural motor functions (sky), physiological functions (red), and cognitive-symbolic (violet) nodes. The blue and red edges represent the positive and negative node correlations based on the functions' polarity relationship. The width of the edges reflects the strength of the correlation.

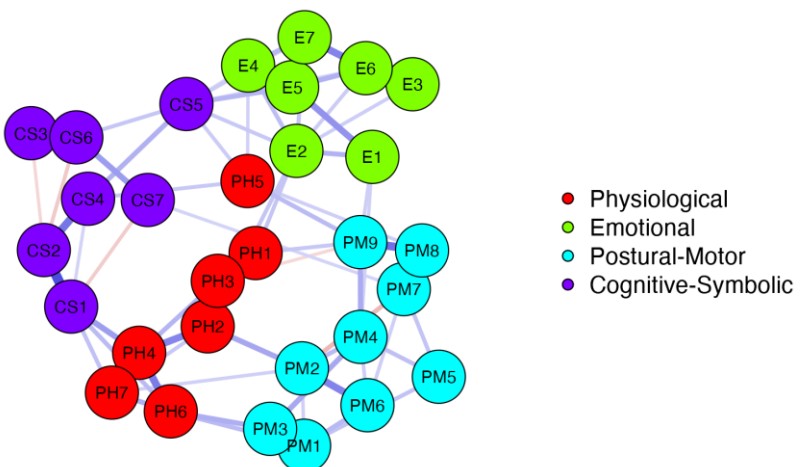

**Figure 1.** Network analysis with estimated partial correlation network factor model for the sixteen BEsS. Number of nodes = 30; number of non-zero edges = 69/435; sparsity = 0.841.

Furthermore, the centrality plot in Figure 2 indicates the position of the nodes in the network analysis. The closeness expresses the sum of the linear distances, the strength measures the sum of the absolute values of the edges per node, and the betweenness shows the sum of the nearness among the nodes [28]. In the centrality plot, to highlight the functions with a significant expected influence, we selected a threshold of 1. The functions PH4, E5, E2, and CS2 had a high expected influence on betweenness, while PH4 had a high expected influence on closeness, and PH4 and E5 had a high expected influence on strength.

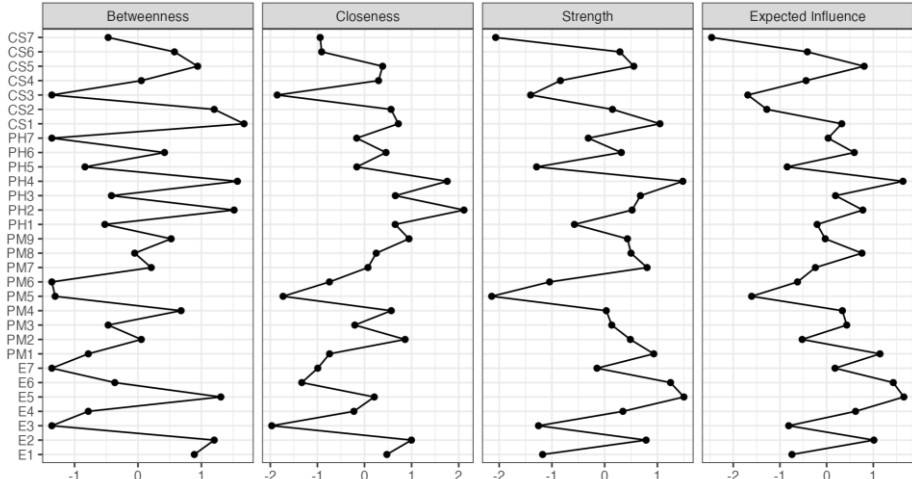

**Figure 2.** Centrality plot illustrating closeness, strength, and expected influence of conditional relations.

Tables 4 and 5 list the means and standard deviations for the thirty functions in any BES. Some of the physiological and postural motor function values have a mean and

standard deviation range from 1 to 3.99 and are marked in bold. These functions accurately represent the values for BEsS 2, 3, 4, 5, 10, and 11. The S.D. values highlighted in red cross the mean of 4 and are considered to be a limit.

**Table 4.** Means and standard deviations results in BEsS from 1 to 8. In the Table, values from 1 to 3.99 are in bold indicating a polarity inversion for that function. The highest standard deviations are highlighted in red, indicating values that cross over the mean of 4. This can be considered a limit in the shift toward a clear polarity.

| Functional | | BES 1 | | BES 2 | | BES 3 | | BES 4 | | BES 5 | | BES 6 | | BES 7 | | BES 8 | |
|---|---|---|---|---|---|---|---|---|---|---|---|---|---|---|---|---|---|
| Plane | | Mn | SD | Mn | SD | Mn | Sd | SD | Mn | SD | Mn | SD | SD | Mn | SD | Mn | SD |
| Physiological | PH1 | 6.31 | 1.00 | 4.86 | 1.64 | 4.17 | 1.53 | 4.01 | 1.11 | 5.18 | 1.28 | 6.04 | 0.89 | 5.90 | 1.11 | 4.97 | 1.54 |
| | PH2 | 6.00 | 1.15 | **2.05** | 1.48 | **1.53** | 1.14 | **1.72** | 1.30 | **3.44** | 1.96 | 5.78 | 1.16 | 5.82 | 1.11 | 4.73 | 1.57 |
| | PH3 | 5.73 | 0.87 | **2.60** | 1.71 | **1.82** | 1.28 | **1.83** | 1.21 | **3.69** | 1.69 | 5.67 | 0.89 | 5.63 | 1.30 | 4.39 | 1.64 |
| | PH4 | 5.76 | 0.90 | **2.43** | 1.49 | **1.73** | 1.19 | **1.97** | 1.48 | **3.67** | 1.72 | 5.80 | 0.87 | 5.68 | 1.00 | 4.75 | 1.37 |
| | PH5 | 5.93 | 0.93 | 6.34 | 0.82 | 5.89 | 1.29 | 6.12 | 0.97 | 6.20 | 0.91 | 5.75 | 0.88 | 5.68 | 1.13 | 5.68 | 0.79 |
| | PH6 | 5.57 | 0.79 | **3.26** | 1.76 | **3.88** | 2.35 | **2.11** | 1.30 | **3.80** | 1.53 | 5.66 | 0.97 | 5.50 | 1.04 | 4.63 | 1.26 |
| | PH7 | 4.70 | 1.33 | **1.93** | 1.09 | **1.66** | 1.04 | **1.89** | 1.12 | **2.95** | 1.40 | 5.25 | 1.08 | 5.09 | 1.18 | 4.36 | 1.37 |
| Postural motor | PM1 | 6.43 | 0.72 | 4.97 | 1.69 | **3.53** | 2.06 | **3.00** | 1.73 | 5.46 | 1.17 | 6.48 | 0.65 | 5.90 | 1.04 | 6.19 | 0.81 |
| | PM2 | 5.72 | 1.06 | **2.10** | 1.19 | **2.02** | 1.37 | **2.11** | 1.41 | **3.40** | 1.66 | 5.30 | 1.33 | 4.89 | 1.62 | 4.90 | 1.65 |
| | PM3 | 6.51 | 0.70 | 5.08 | 1.67 | 4.01 | 2.07 | **2.61** | 1.68 | 5.65 | 1.09 | 6.51 | 0.65 | 6.14 | 0.98 | 6.07 | 0.91 |
| | PM4 | 6.20 | 0.95 | 5.01 | 1.28 | 4.49 | 1.39 | 4.39 | 1.00 | 5.50 | 0.89 | 5.99 | 1.23 | 5.79 | 1.19 | 5.79 | 1.19 |
| | PM5 | 5.71 | 1.13 | 5.27 | 1.58 | 4.00 | 2.07 | 4.63 | 1.94 | 5.74 | 0.96 | 5.08 | 1.45 | 4.64 | 1.57 | 5.39 | 1.37 |
| | PM6 | 5.48 | 1.31 | 5.12 | 1.84 | **2.28** | 1.56 | **2.44** | 1.54 | **3.38** | 1.73 | 5.01 | 1.62 | 4.59 | 1.54 | 5.37 | 1.37 |
| | PM7 | 5.16 | 1.48 | 6.12 | 1.23 | 5.86 | 1.39 | 5.98 | 1.09 | 6.15 | 0.92 | 4.79 | 1.59 | 4.62 | 1.67 | 5.68 | 1.11 |
| | PM8 | 5.71 | 1.16 | 6.45 | 0.99 | 6.27 | 1.25 | 6.59 | 0.69 | 6.43 | 0.74 | 5.20 | 1.35 | 5.04 | 1.44 | 5.81 | 1.11 |
| | PM9 | 6.35 | 0.90 | 6.53 | 0.84 | 6.51 | 0.91 | 6.51 | 0.73 | 6.44 | 0.71 | 6.02 | 1.08 | 5.87 | 1.22 | 6.19 | 0.82 |
| Cognitive Symbolic revised | CS1 | 5.49 | 1.20 | **2.58** | 1.72 | **1.78** | 1.26 | **2.03** | 1.54 | **3.97** | 1.90 | 6.09 | 0.84 | 5.34 | 1.34 | **3.10** | 1.56 |
| | CS2 | 4.72 | 1.27 | 4.20 | 1.32 | **3.71** | 1.55 | **3.54** | 1.47 | 4.61 | 1.46 | 5.75 | 1.00 | 4.69 | 1.41 | **3.40** | 1.54 |
| | CS3 | 4.68 | 1.52 | 4.10 | 1.44 | 4.72 | 1.36 | 4.71 | 1.34 | 4.68 | 1.67 | **3.47** | 2.00 | 4.11 | 1.59 | 4.79 | 1.24 |
| | CS4 | 5.12 | 1.18 | 5.25 | 1.27 | 4.90 | 1.38 | 4.98 | 1.38 | 5.48 | 1.07 | 5.36 | 1.00 | 4.94 | 1.17 | 4.22 | 1.53 |
| | CS5 | 6.37 | 0.78 | 6.48 | 0.80 | 6.34 | 0.86 | 6.33 | 0.86 | 6.55 | 0.65 | 5.89 | 1.16 | 5.65 | 1.16 | 6.00 | 0.81 |
| | CS6 | 5.62 | 1.23 | 4.94 | 1.01 | 4.83 | 1.06 | 4.95 | 1.21 | 5.68 | 1.10 | 4.64 | 1.86 | 5.16 | 1.25 | 5.12 | 1.12 |
| | CS7 | 4.96 | 1.74 | 4.95 | 1.80 | 5.56 | 1.39 | 5.81 | 1.23 | 5.35 | 1.63 | 4.98 | 1.77 | 4.85 | 1.62 | 5.50 | 1.19 |
| Emotional | E1 | 6.52 | 0.68 | 6.70 | 0.59 | 6.61 | 0.79 | 6.50 | 0.72 | 6.45 | 0.56 | 6.01 | 1.11 | 6.32 | 0.79 | 6.42 | 0.76 |
| | E2 | 6.60 | 0.65 | 5.53 | 1.06 | 4.76 | 0.91 | 5.13 | 0.74 | 6.05 | 0.78 | 5.62 | 1.21 | 6.05 | 0.95 | 6.15 | 0.77 |
| | E3 | 5.50 | 1.49 | 6.49 | 0.99 | 6.68 | 0.72 | 6.73 | 0.52 | 6.38 | 0.77 | 4.68 | 1.71 | 4.55 | 1.78 | 5.84 | 1.17 |
| | E4 | 6.25 | 0.90 | 6.34 | 0.93 | 6.22 | 1.00 | 6.36 | 0.77 | 6.40 | 0.66 | 6.08 | 0.92 | 5.89 | 1.06 | 6.19 | 0.92 |
| | E5 | 6.63 | 0.55 | 6.78 | 0.52 | 6.72 | 0.55 | 6.69 | 0.61 | 6.64 | 0.53 | 6.14 | 0.96 | 6.23 | 0.83 | 6.46 | 0.62 |
| | E6 | 5.94 | 0.93 | 6.67 | 0.63 | 6.80 | 0.47 | 6.78 | 0.47 | 6.47 | 0.67 | 4.88 | 1.60 | 5.00 | 1.46 | 6.20 | 0.76 |
| | E7 | 6.43 | 0.81 | 6.82 | 0.51 | 6.38 | 0.95 | 6.67 | 0.62 | 6.80 | 0.46 | 5.75 | 1.12 | 5.76 | 1.15 | 6.16 | 0.81 |

**Table 5.** Means and standard deviations results in BEsS from 9 to 16. In the Table, values from 1 to 3.99 are in bold indicating a polarity inversion for that function. The highest standard deviations are highlighted in red, indicating values that cross over the mean of 4. This can be considered a limit in the shift toward a clear polarity.

| Functional | | BES 9 | | BES 10 | | BES 11 | | BES 12 | | BES 13 | | BES 14 | | BES 15 | | BES 16 | |
|---|---|---|---|---|---|---|---|---|---|---|---|---|---|---|---|---|---|
| Plane | | Mn | SD | Mn | SD | Mn | Mn | SD | Mn | SD | Mn | SD | SD | Mn | SD | Mn | SD |
| Physiological | PH1 | 6.09 | 1.13 | **2.95** | 1.61 | **3.46** | 1.35 | 4.48 | 1.73 | 5.94 | 1.16 | 5.87 | 1.40 | 4.21 | 1.58 | 5.74 | 1.60 |
| | PH2 | 5.91 | 1.26 | **2.41** | 1.67 | **1.79** | 1.44 | 4.76 | 1.77 | 5.22 | 1.43 | 4.98 | 1.67 | 4.30 | 2.02 | 5.86 | 1.13 |
| | PH3 | 5.57 | 1.46 | **1.91** | 1.29 | **1.75** | 1.38 | 4.07 | 1.89 | 5.17 | 1.41 | 5.13 | 1.49 | 3.51 | 1.68 | 6.03 | 0.94 |
| | PH4 | 5.67 | 1.06 | **2.48** | 1.43 | **2.15** | 1.56 | 5.19 | 1.41 | 5.42 | 1.22 | 5.35 | 1.33 | 4.21 | 1.87 | 6.03 | 0.93 |
| | PH5 | 5.77 | 0.88 | 5.33 | 1.10 | 5.20 | 1.59 | 5.52 | 1.03 | 5.66 | 0.98 | 5.94 | 1.09 | 5.80 | 1.11 | 5.17 | 1.58 |
| | PH6 | 5.48 | 0.92 | **2.94** | 1.47 | **2.23** | 1.45 | 5.17 | 1.19 | 5.22 | 1.14 | 5.48 | 1.06 | 4.27 | 1.72 | 5.94 | 1.00 |
| | PH7 | 4.99 | 1.20 | **2.45** | 1.38 | **2.10** | 1.45 | 4.40 | 1.58 | 4.62 | 1.47 | 5.06 | 1.40 | **3.66** | 1.80 | 5.73 | 1.07 |

**Table 5.** *Cont.*

| Functional | | BES 9 | | BES 10 | | BES 11 | | BES 12 | | BES 13 | | BES 14 | | BES 15 | | BES 16 | |
|---|---|---|---|---|---|---|---|---|---|---|---|---|---|---|---|---|---|
| Plane | | Mn | SD | Mn | SD | Mn | Mn | SD | Mn | SD | Mn | SD | SD | Mn | SD | Mn | SD |
| Postural motor | PM1 | 6.24 | 0.88 | **3.65** | 1.78 | **3.06** | 1.86 | 5.90 | 1.01 | 5.45 | 1.14 | 5.89 | 0.95 | 5.11 | 1.50 | 5.43 | 1.43 |
| | PM2 | 5.67 | 1.21 | **2.50** | 1.43 | **2.15** | 1.44 | 4.94 | 1.55 | 4.56 | 1.47 | 4.48 | 1.73 | **3.70** | 1.86 | 5.15 | 1.62 |
| | PM3 | 6.44 | 0.77 | **3.49** | 1.76 | **3.00** | 1.89 | 6.14 | 0.90 | 5.82 | 1.05 | 6.09 | 0.95 | 5.29 | 1.57 | 6.00 | 1.21 |
| | PM4 | 6.04 | 1.10 | 4.52 | 1.24 | 4.16 | 1.24 | 5.69 | 1.15 | 5.44 | 1.13 | 5.81 | 1.12 | 5.33 | 1.18 | 5.53 | 1.39 |
| | PM5 | 5.40 | 1.28 | 4.70 | 1.66 | 4.75 | 1.83 | 5.48 | 1.23 | 5.11 | 1.25 | 5.52 | 1.36 | 5.64 | 1.14 | 4.08 | 1.70 |
| | PM6 | 5.07 | 1.48 | **2.56** | 1.57 | **2.14** | 1.45 | 4.37 | 1.78 | 4.10 | 1.62 | 4.42 | 1.81 | **3.60** | 1.89 | 4.60 | 1.69 |
| | PM7 | 4.92 | 1.54 | 5.12 | 1.65 | 5.61 | 1.53 | 4.99 | 1.53 | 4.76 | 1.54 | 5.33 | 1.55 | 5.64 | 1.19 | **3.38** | 1.74 |
| | PM8 | 5.37 | 1.24 | 5.57 | 1.43 | 5.92 | 1.45 | 5.35 | 1.27 | 5.02 | 1.41 | 5.63 | 1.49 | 5.99 | 0.95 | **3.52** | 1.70 |
| | PM9 | 6.21 | 0.93 | 5.98 | 1.23 | 5.99 | 1.20 | 5.77 | 1.06 | 5.74 | 0.99 | 5.90 | 1.21 | 6.14 | 0.79 | 4.67 | 1.67 |
| Cognitive symbolic | CS1 | 5.46 | 1.45 | **3.17** | 1.86 | **2.43** | 1.84 | 5.96 | 1.11 | 5.75 | 1.11 | 5.91 | 1.16 | 4.57 | 2.11 | 6.09 | 0.89 |
| | CS2 | 4.94 | 1.36 | **3.72** | 1.48 | **3.53** | 1.73 | 5.50 | 1.28 | 5.45 | 1.31 | 5.44 | 1.49 | 4.60 | 1.70 | 5.19 | 1.68 |
| | CS3 | 4.18 | 1.65 | 4.34 | 1.33 | 4.65 | 1.68 | 4.49 | 1.94 | **3.62** | 1.83 | **3.85** | 2.03 | 4.33 | 1.77 | **3.60** | 1.74 |
| | CS4 | 5.39 | 1.04 | 4.34 | 1.33 | 4.64 | 1.57 | 5.75 | 1.07 | 5.64 | 1.10 | 5.49 | 1.29 | 5.24 | 1.26 | 4.90 | 1.51 |
| | CS5 | 6.07 | 0.85 | 5.61 | 0.97 | 5.97 | 1.10 | 6.04 | 0.88 | 5.46 | 1.42 | 5.76 | 1.41 | 6.30 | 0.72 | **3.66** | 1.77 |
| | CS6 | 4.91 | 1.52 | 4.44 | 1.16 | 4.79 | 1.37 | 4.97 | 1.81 | 4.10 | 1.85 | 4.58 | 1.91 | 4.76 | 1.69 | **3.64** | 1.76 |
| | CS7 | 4.79 | 1.65 | 4.94 | 1.64 | 5.24 | 1.72 | 4.42 | 2.00 | 4.89 | 1.74 | 5.08 | 1.81 | 4.91 | 1.96 | 4.28 | 1.99 |
| Emotional | E1 | 6.36 | 0.93 | 6.34 | 0.81 | 6.33 | 0.78 | 6.26 | 0.83 | 6.13 | 0.97 | 5.84 | 1.49 | 6.44 | 0.70 | 5.70 | 1.48 |
| | E2 | 6.26 | 0.85 | 4.82 | 0.96 | 4.98 | 0.81 | 5.46 | 0.92 | 5.35 | 1.24 | 4.70 | 1.73 | 5.99 | 0.73 | **3.74** | 1.61 |
| | E3 | 5.32 | 1.58 | 6.24 | 1.11 | 6.63 | 0.60 | 5.89 | 1.07 | 5.10 | 1.58 | 4.94 | 1.84 | 6.48 | 0.71 | **2.99** | 1.72 |
| | E4 | 6.10 | 0.97 | 5.73 | 1.08 | 6.15 | 0.96 | 5.99 | 0.89 | 5.56 | 1.39 | 5.83 | 1.45 | 6.38 | 0.64 | **3.53** | 1.91 |
| | E5 | 6.31 | 0.81 | 6.15 | 0.87 | 6.51 | 0.66 | 6.21 | 0.75 | 5.76 | 1.24 | 5.99 | 1.35 | 6.58 | 0.60 | **3.91** | 1.90 |
| | E6 | 5.74 | 1.07 | 6.49 | 0.72 | 6.73 | 0.52 | 6.03 | 0.88 | 5.17 | 1.49 | 5.42 | 1.52 | 6.52 | 0.56 | **2.99** | 1.65 |
| | E7 | 5.94 | 1.03 | 5.88 | 0.93 | 6.36 | 0.80 | 5.89 | 0.88 | 5.58 | 1.18 | 6.09 | 1.29 | 6.55 | 0.64 | **3.68** | 1.67 |

We observed a shift towards the right polarity in the following functions: PH2, PH3, PH4, PH6, and PH7 in BEsS 2, 3, 4, 10, and 11, as well as in function CS1 in BEsS 2, 3, and 4. Additionally, for the functions PH1 in BES 10; PH6 in BEsS 10 and 11; and PM2 in BEsS 2, 3, 4, 10, and 11, as well as PM6 in BEsS 3, 4, 10, and 11, we also noted a shift towards the right polarity. Furthermore, there was a high standard deviation in the functions PH2, PH3, PH4, PH6, and PH7 in BES 5; PH1 in BES 11; CS1 in BES 5; and PM2 in BES 5. This can cause inaccurate polarity marking, which overlaps with the value of 4, which is considered neutral in the Osgood scale. The results of the ANOVA and post hoc Bonferroni and Tukey tests indicated that there were significant differences among the mean values of BEsS 2, 3, 4, 10, and 11. Hence, we separated the BEsS into two groups. The first group comprised BesS 1-Vitality, 6-Strength, 7-Joyous Aggression, 9-Playing, and 16-Opposition. The second group included BesS 2-Contact, 3-Letting Go, 4-Being Held, 10-Just Being, and 11-Calmness. Furthermore, a statistical analysis using Welch's $t$-test revealed significant differences between the two groups in all the functions except CS6 and PH5. We also confirmed the significant difference between the two groups of BEsS through a giant d-Cohen effect size (d > 1.0) in functions PH1, PH2, PH3, PH4, PH6, PH7, PM1, PM2, PM3, PM4, PM6, E3, E6, and CS1 (Table 6).

**Table 6.** Differences between Group 1 and Group 2. Independent Welch's $t$-test ($p < 0.05$). Cohen's d-effect size. In italics, giant d-Cohen (d > 1.0) effect size with functions in decretive order. In the table below, functions with Cohen's d values greater than 1.00 are italicized.

| Functions | t | df | p | Cohen's d | SE Cohen's d |
|---|---|---|---|---|---|
| *PH2* | *84.163* | *2974.171* | *<0.001* | *3.057* | *0.068* |
| *PH3* | *82.440* | *2972.786* | *<0.001* | *2.994* | *0.067* |
| *PH4* | *81.522* | *2757.524* | *<0.001* | *2.946* | *0.066* |
| *PH7* | *67.721* | *2995.618* | *<0.001* | *2.470* | *0.059* |
| *PM2* | *64.481* | *2967.177* | *<0.001* | *2.355* | *0.057* |

**Table 6.** *Cont.*

| Functions | t | df | p | Cohen's d | SE Cohen's d |
|---|---|---|---|---|---|
| *CS1* | *59.734* | *2821.124* | *<0.001* | *2.161* | *0.055* |
| *PH6* | *57.322* | *2492.794* | *<0.001* | *2.063* | *0.053* |
| *PM3* | *51.455* | *2247.437* | *<0.001* | *1.846* | *0.050* |
| *PM1* | *46.431* | *2468.414* | *<0.001* | *1.671* | *0.048* |
| *PH1* | *45.124* | *2919.933* | *<0.001* | *1.636* | *0.048* |
| *PM6* | *35.717* | *2985.076* | *<0.001* | *1.298* | *0.044* |
| *E6* | *−36.196* | *1711.840* | *<0.001* | *−1.349* | *0.044* |
| *E3* | *−35.244* | *1904.732* | *<0.001* | *−1.309* | *0.044* |
| *PM4* | *33.088* | *2998.262* | *<0.001* | *1.207* | *0.043* |
| CS2 | 24.666 | 3005.996 | <0.001 | 0.898 | 0.040 |
| PM8 | −20.722 | 2776.927 | <0.001 | −0.760 | 0.039 |
| E7 | −19.360 | 2160.835 | <0.001 | −0.717 | 0.039 |
| PM7 | −18.918 | 2820.862 | <0.001 | −0.694 | 0.039 |
| E2 | 14.098 | 2329.513 | <0.001 | 0.521 | 0.038 |
| E5 | −16.004 | 1981.467 | <0.001 | −0.594 | 0.038 |
| PM5 | 6.020 | 2983.332 | <0.001 | 0.219 | 0.037 |
| CS6 | 0.554 | 2558.038 | 0.580 | 0.020 | 0.037 |
| E4 | −10.971 | 2340.096 | <0.001 | −0.405 | 0.037 |
| CS5 | −10.978 | 2432.055 | <0.001 | −0.405 | 0.037 |
| PM9 | −8.689 | 2763.780 | <0.001 | −0.319 | 0.037 |
| CS3 | −8.146 | 2799.617 | <0.001 | −0.299 | 0.037 |
| E1 | −7.779 | 2538.061 | <0.001 | −0.286 | 0.037 |
| CS7 | −7.755 | 2913.996 | <0.001 | −0.284 | 0.037 |
| PH5 | −0.075 | 2992.725 | 0.940 | −0.003 | 0.037 |

## 5. Discussion

Our research aims to explore the Basic Experience of the Self, which is a functional developmental theory construct. We designed the BAF and used the Osgood semantic scale to evaluate the level of the functions' polarity in the spontaneous behaviour of infants between the ages of zero and three years. The infants' spontaneous activity was assessed using data from a previous study [25]. In this article, we discuss the results of sixteen BEsS studies.

The BAF demonstrates good and acceptable internal consistency for any BES. However, the Cronbach's alpha value for the BEsS Letting Go and Being Held is questionable. This may be due to the limited data collected and the similarity in functional characteristics between the BEsS. Although only two of the sixteen BEsS showed the BAF's questionable internal consistency, it is important to exercise caution when using the BAF to assess Letting Go and Being Held. Nevertheless, we plan to extend the BAF assessment by including older children for the same BEsS study.

Our findings support the hypotheses stated in the functional developmental theory. Each function can be measured on a spectrum ranging between its opposite polarities. Based on the grading of the functions, every BES displays a typical function configuration that aligns with the theory. Our analysis, which is consistent with the functional developmental theory, identified four functional planes that classify the 30 functions. In our research, we identified consistent relationships between the functions within each BES configuration and when comparing the BEsS. The network analysis of all sixteen BEsS depicts a clear connection among the functions that categorise each domain, as described in the theory. Each function's domain classifies a group based on its functioning and shows positive or negative conditional associations among the functions that are also in the same domain. This association might be typical in each BES. Ongoing elaborations are being made to further understand whether the associations are typical in each BES. Figure 1 displays the results of the network analysis, where the clear grouping is documented. The red edges highlight negative correlations among the functions, such as those between PM2 (fast/slow movements) and PM7 (soft/jerky movements). The analysis consistently shows

that slower movements tend to be smoother, while faster movements are less smooth. This finding is consistent with functional theory and aligns with observable behaviour in infants. Understanding the interrelationships between the various functions can be a challenging task, especially when there are multiple function relations at play. To gain a better understanding of these relationships, further investigation and analysis are required. We intend to explore alternative network analysis models and to analyse any factors that may contribute to the configuration structure or specificity of the BEsS. It is important to note that our study is still ongoing, and we will continue to update our findings.

We observed a consistent inversion in the polarity mark in most of the physiological and some of the postural motor functions. The values ranged from 3.99 to 1 in the following BEsS: 2-Contact, 3-Letting Go, 4-Being Held, 10-Just Being, and 11-Calmness. In relation to other BEsS, we previously observed a significant intercorrelation function result in PH4 in the centrality plot of the network analysis. The statistical analysis, including ANOVA, the post hoc tests, and the d-Cohen effect size, confirmed the significant differences. These results indicate that the physiological and postural motor functions of the first group clearly show a left-side polarity mark for the BEsS. Specifically, the BEsS Vitality, Strength, Joyous Aggression, Playing, and Opposition are characterised by an open voice, excitation, noise, sympathicotonia, muscle hypertonia, and thoracic breathing, along with postural motor mobility, fast movements, activeness, nimbleness, and big movements. Certain physical characteristics can be observed when shifting to right-side polarity for the BEsS Contact, Letting Go, Being Held, Just Being, and Calmness. These include a closed voice, calmness, relaxed muscles, diaphragmatic breathing, and immobility, as well as slow, passive, clumsy, and small movements. The functions of attention and loosening are classified into the cognitive-symbolic domain. According to the findings, we suggest the potential to transform the meaning of words from a cognitive-symbolic perspective to a physiological or postural motor perspective. This implies that their semantic significance could be impacted by bodily activation.

Our argument is that positive values in physiological and postural motor functions indicate observable arousal in the activity of children in Group 1. Conversely, the reversal of polarity in physiological and postural motor functions indicates a decrease in arousal in Group 2. These observations are consistent with the existence of a 'cerebral energy' that regulates the biobehavioural adaptation of infants in the last trimester of prenatal age, which is responsible for the organisation of circadian rhythm and heart rate variability [29–31].

It is crucial to have a clear understanding of the meaning and limitations of BAF while evaluating and measuring an infant's BES functions. Our study graded the functions mentioned in Rispoli's publication [18] based on the semantic connotation derived from their theoretical basis. The semantic connotation of most of the implied functions can be comprehended based on our experiences and specific training in functionalist psychotherapy. Furthermore, it is difficult to determine physiological functions such as "sympathicotonia", "excitation", or "thoracic breathing" without objective measurement tools to analyse heart rate or epidermal impedance. The only way we can speculate is via our perception of body characteristics when energy is released and when the body moves. We suggest that this area deserves more research. It is important to develop an experimental framework to document the fact that functions like excitation or sympathicotonia are characterised by cardio-respiratory activation. This can be demonstrated by a higher mean heart rate or an increased rate in the respiratory cycle, combined with a small increase in respiratory acts. Furthermore, appropriate tools could evaluate cognitive-symbolic functions when the study is extended to older children.

## 6. Conclusions

The study of psychometrics should relate theoretical constructs to the complexity of child development. However, clinical practice has only focused on partial aspects of neurodevelopment, resulting in incomplete assessments of development. Developmental science has emerged to guide research in social, psychological, and biobehavioural sciences.

However, current clinical assessment methods for infant development and early diagnosis could be improved.

To make accurate clinical neurodevelopmental observations, it is necessary to take a comprehensive approach that encompasses physiological, motor, cognitive, and emotional development. Based on neo-functionalist developmental theory, the BAF is a psychometric tool designed to assess infants' functioning complexity during daily activities. BAF's internal consistency suggests that each item is significant, but semantic modifications may be needed. This research highlights some important aspects of developmental theory that help in validating the construct. The study enables us to assess the polarity of the functions that are observed in infant activity, to categorise them into four domains based on their significance, and to identify a BES by analysing the presence of an arousal activity. The results consistently showed that positive polarity in physiological and postural motor functions is observed in the presence of an arousal activity, while the same functions exhibit an inversion in the case of a loosening arousal activity.

It is important to note that even though we have collected a large number of forms for this study, we must acknowledge certain limitations. Obtaining a consistent number of videos for each BES required a lot of effort. Although we carefully selected the videos to highlight the distinct features of each BES, there is a chance of mixed and overlapping influences. Therefore, it is crucial to consider the semantic meaning of each function to eliminate any biases that may affect the characterisation of the specific function. However, our positive results motivate us to continue with our research process.

**Author Contributions:** Conceptualisation, F.D., M.B., R.G., P.F. and P.M.; methodology, P.M.; investigation, F.D., M.B. and R.G.; data curation, F.D.; writing—original draft preparation, F.D.; writing—review and editing, F.D., M.B., R.G., P.F. and P.M.; supervision, P.M. All authors have read and agreed to the published version of the manuscript.

**Funding:** This research received no external funding.

**Institutional Review Board Statement:** The study was conducted in accordance with the Declaration of Helsinki and approved by the Institutional Review Board of Functional Psychotherapy Centre, Catania, Italy. Protocol code 39/2022. Date of approval: 14 November 2022.

**Informed Consent Statement:** Informed consent was obtained from all subjects involved in the study.

**Data Availability Statement:** Data will be available upon a reasonable request to the corresponding author.

**Acknowledgments:** To students at the European School of Psychotherapy in Catania (Italy) and all the expert functional psychotherapists participating in the BAF video analysis. Families freely provided clips of infants that allowed them to acquire a large amount of data.

**Conflicts of Interest:** Authors Paola Fecarotta, Marta Blandini, and Raffaele Gueli were employed by the Company Istituto di Psicoterapia Funzionale–Centro Wilhelm Reich. 95124. Catania (Italy). The remaining authors declare that the research was conducted in the absence of any commercial or financial relationships that could be construed as a potential conflict of interest.

**Appendix A**

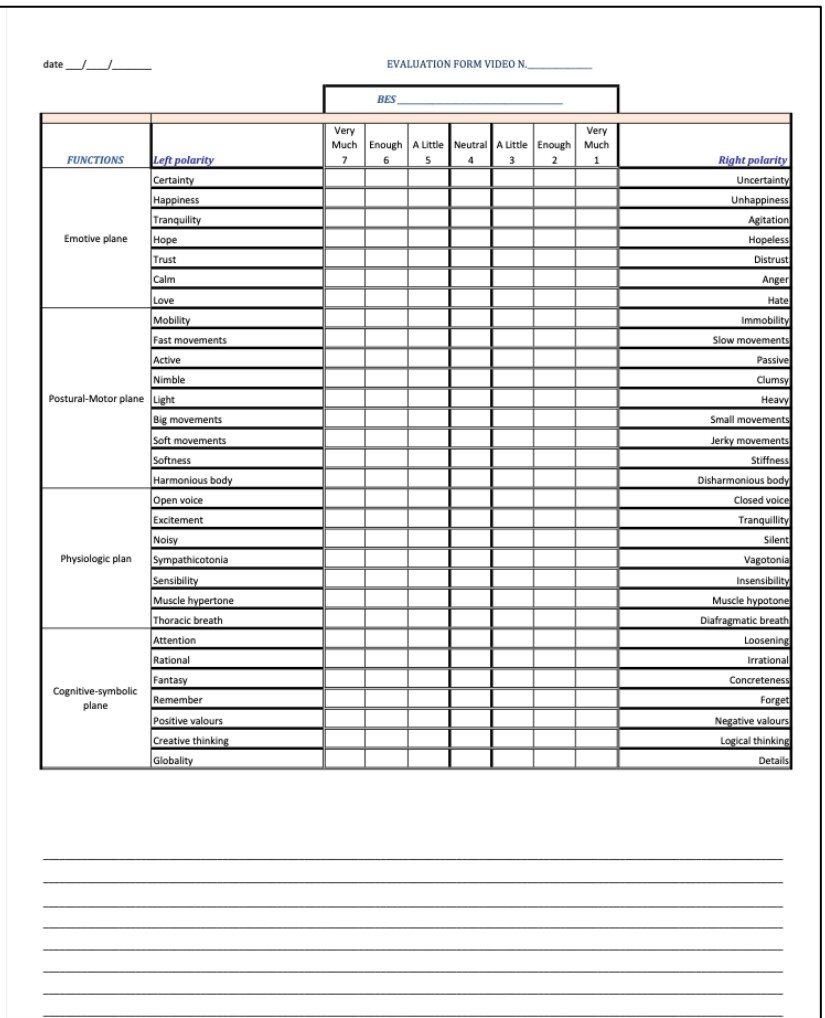

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
