# Peer review of "On the Construct of Functional Psychology’s Developmental Theory: Basic Experiences of the Self (BEsS)"

_ejihpe, doi:10.3390/ejihpe13120198_

Round 1

Reviewer 1 Report

Comments and Suggestions for Authors

1.1.  I miss a section in which it is specified which is the hypothesis (or hypotheses) that they want to prove. In lines 17, 30, 180 and 195 the word hypothesis appears mentioned in passing, without specifying (for instance in the classic format If... then) what is the nexus detected between A & B (antecedent and consequent, predictor – criterion). In line 17 (in the abstract) the hypothesis is linked to BEsS characteristics, in line 30 is linked to DOHaD, in 180 to physiological activation, and in 195 to functions through network analysis. Please, clarify how many and please, keep in mind the distinction Hypothesis (the verb is typically used to express the nexus in the present simple tense) versus Predictions (expressed with the verb in the future simple tense). In science, it's reasonable to base various predictions on a central and explained hypothesis.

2.2.    In line 54 Boadella is mentioned but I miss an entry in the references’ section. Again, in line 69 Reich is mentioned and an entry in the reference section is recommended. If Wilheim Reich is the author mentioned, then the connection with the Chicago School should be elucidated. How Reich's psychoanalytic practice relates to findings detected during the first thousand days of a baby? Nothing appears in the manuscript. The nexus between Dewey and functional psychology is a part of the history of psychology. But not the nexus between Reich and Dewey.

33. In English the term 'self' is used, both, in everyday and technical contexts and refers to oneself as an object. The same linguistic aspect emerges in “Moi”. How do you introduce the notion of self in your scales in Italian language? The news that come to me is that it is a linguistic whirlwind.

44. Table 3 is confusing due to the fact that lines 234 to 235 display the same columns and rows as lines 236 to 237, but the numbers differ. My suggestion is that this table n.3 should be transformed, for example, into table 3a and 3b, and the content commented separately first, and then combined.

55.  If this is a new section, a blank line should be inserted between lines 244 and 245. However in line 186 there is already the heading: 4. Results and in line 258 the heading 5. Discussion appears.

Comments on the Quality of English Language

Although I believe the English is correct, but it's not my native language. Therefore, I suggest having a native review what's written.

Author Response

Please find our responses to your comments in the attached file.

Reviewer 2 Report

Comments and Suggestions for Authors

Report assessment article entitled On the Construct of Functionalism Psychology’s Developmen-2 tal Theory: Basic Experiences of the Self (BEsS).

The article provides knowledge in relation to child development in the first years of life taking into account the interrelation and interaction between different functions and domains.

Some areas for improvement in the article are as follows:

-          The abstract could be written according to the IMRD format, this would help the subsequent readability of the article.

-          The objective of the work and the hypotheses should be included at the end of the introduction. This would help to understand the subsequent sections more clearly.

-          In the method section, a description of the sample could be included, in relation to the age and sex of the children who were recorded in the videos. Whether neurodevelopmental disorders have been ruled out, as well as socio-familial characteristics that may be relevant.  In addition, the environmental conditions surrounding the recording of the videos could be described in more detail.

-          The authors note that the study was approved by the ethics committee on 13 October 2023. According to this, the study started before the corresponding report was obtained.

-          Line 179 Our previous research discovered that… could you please provide the references of these studies.

-          There is an error in the numbering of the tables. This should be checked.

-          The titles of the tables can be modified. It is necessary to differentiate between the title itself and notes that help to interpret the table. The latter should go at the end of the table by specifying Note:.......

-          Table 3 has errors in the columns. For example in column BES 3 it says Mn and Mn, it should say Mn and SD. Check this in all columns.

-          Some of the tables do not follow APA standards.

-          In the discussion section it should be mentioned whether the hypotheses previously stated in the study are fulfilled. In addition, a paragraph on practical implications and future studies could be added.

Author Response

(The authors gave the same response as above.)

Reviewer 3 Report

Comments and Suggestions for Authors

Here are my suggestions:

1. Introduction

  • Lines 51-70: The section discussing the historical context and development of Neo-functionalism, starting from Pierre Janet to Reich's formulations, is informative but somewhat condensed. The authors might want to elaborate on how these historical theories have specifically influenced the contemporary understanding and framing of Neo-functionalism. This could include discussing key theoretical shifts or additions that have led to the current conceptualization of the Self within this framework.

2. Rationale of the study

  • Lines 109-136: While the manuscript lists various Basic Experiences of the Self (BEsS), there seems to be a missing link between these descriptions and the larger theoretical framework of Neo-functionalism. The authors might want to elucidate further how each of these BEsS fits into the developmental theory of Neo-functionalism. Explaining how these experiences collectively contribute to the development and functioning of the Self according to this theoretical perspective can enhance the manuscript's theoretical coherence.

3. Materials and Methods

  • Line 148: The authors mention choosing videos of infants based on planned BES criteria, yet the specifics of these criteria still need to be elaborated upon. They might want to provide more detail about the selection process for these videos. For example, what characteristics or behaviors were they specifically looking for in these videos to categorize them according to different BESs? This would aid in understanding the relevance of the chosen videos to the study's aims.

4. Results

The authors might want to consider expanding on the interpretation and implications of the lower Cronbach's alpha values for the BESs “Letting Go” and “Being Held” in lines 186-189. While the manuscript acknowledges these as "questionable," it would be beneficial to discuss how these values impact the overall reliability of the BAF (Basic Experiences of the Self) instrument and any considerations or revisions that may be necessary for future research.

In the section starting from line 207, the authors might want to provide a more detailed explanation of how the network analysis findings support the functional theory's classification into four discrete domains. While the results are presented, there seems to be a lack of in-depth discussion connecting these results back to the theoretical framework of functionalism psychology's developmental theory. Elaborating on this connection would strengthen the theoretical implications of the study.

5. Discussion and 6. Conclusion

  • Lines 270-274: While the manuscript states that "no other analogue research has been published to compare this data," the authors might want to explore and discuss how their findings align or diverge from theoretical expectations or related constructs in developmental psychology. This comparison could be theoretical rather than empirical, providing a broader context for their findings.

  • Lines 305-309 and 329-335: The authors might want to expand on the future research directions by specifically suggesting how subsequent studies could address the limitations, such as using objective measurement tools (like heart rate or epidermal impedance). Furthermore, mentioning the challenges in obtaining a consistent number of videos for each BES and the potential for mixed and overlapping influences warrants a more detailed discussion on how this might impact the study's findings and generalizability.

Author Response

(The authors gave the same response as above.)
